# Multi-trait association analysis reveals shared genetic loci between Alzheimer's disease and cardiovascular traits

Fotios Koskeridis [1,2,3] ✉, Nurun Fancy [3,4], Pei Fang Tan[5,6], Devendra Meena[1], Evangelos Evangelou[1,2], Paul Elliott [1,3], Dennis Wang [5,6,7], Paul M. Matthews [3,4], Abbas Dehghan [1,3] & Ioanna Tzoulaki [1,3,8]

Several cardiovascular traits and diseases co-occur with Alzheimer's disease. We mapped their shared genetic architecture using multi-trait genome-wide association studies. Subsequent fine-mapping and colocalisation highlighted 16 genetic loci associated with both Alzheimer's and cardiovascular diseases. We prioritised rs11786896, which colocalised with Alzheimer's disease, atrial fibrillation and expression of *PLEC* in the heart left ventricle, and rs7529220, which colocalised with Alzheimer's disease, atrial fibrillation and expression of *C1Q* family genes. Single-cell RNA-sequencing data, co-expression network and protein-protein interaction analyses provided evidence for different mechanisms of *PLEC*, which is upregulated in left ventricular endothelium and cardiomyocytes with heart failure and in brain astrocytes with Alzheimer's disease. Similar common mechanisms are implicated for *C1Q* in heart macrophages with heart failure and in brain microglia with Alzheimer's disease. These findings highlight inflammatory and pleomorphic risk determinants for the co-occurrence of Alzheimer's and cardiovascular diseases and suggest PLEC, C1Q and their interacting proteins as potential therapeutic targets.

Alzheimer's disease (AD), the most common cause of dementia, is a leading health challenge for our times. More than 55 million people worldwide were estimated to be living with dementia in 2020 with 60%-70% of them being AD cases[1,2]. AD has been considered a brain-specific disease whose primary pathology is confined to the brain. However, accumulating evidence suggests mechanistic links between a wide range of cardiovascular (CV) abnormalities and AD[3–5]. Epidemiological studies and experimental data have shown consistent associations between manifestations of clinical CV diseases such as coronary artery disease (CAD), atrial fibrillation (AF) and stroke, with higher risk of AD[6–8]. Several hypotheses have been proposed to explain this. Indeed, atherosclerosis, the main underlying cause of CV diseases, also has profound consequences on the cerebrovascular system. These include reduced blood flow and potential vascular damage in the brain, impaired cerebral perfusion, and associations with inflammation and oxidative stress, all of which are factors that can contribute to neurodegenerative pathology and increase the risk of AD[9]. Beyond atherosclerosis, other hemodynamic effects associated with hypertension, arteriosclerosis and subsequent aortic stiffening have been associated with cerebrovascular damage and cognitive function, potentially accelerating the onset and progression of AD[10].

[1]Department of Epidemiology and Biostatistics, School of Public Health, Imperial College London, London, UK. [2]Department of Hygiene and Epidemiology, University of Ioannina Medical School, Ioannina, Greece. [3]UK Dementia Research Institute, Imperial College London, London, UK. [4]Department of Brain Sciences, Imperial College London, London, UK. [5]Institute for Human Development and Potential, Agency for Science, Technology and Research (A*STAR), Singapore, Republic of Singapore. [6]Bioinformatics Institute, Agency for Science, Technology and Research (A*STAR), Singapore, Republic of Singapore. [7]National Heart and Lung Institute, Imperial College London, London, UK. [8]Systems Biology, Biomedical Research Institute of the Academy of Athens, Athens, Greece. ✉e-mail: f.koskeridis@imperial.ac.uk

AD and CV traits also share common genetic determinants[11]. Genome-wide association studies (GWAS) have identified genetic risk loci for both AD and pathological CV traits and identified common genetic factors that may refer to the shared underlying pathways. One example of such genes is apolipoprotein E (*APOE*), which encodes a lipid-transport protein involved in cholesterol metabolism[12], that is the strongest genetic risk factor for AD[13,14] and a risk factor for adverse CV traits, including CAD[15] and myocardial infarction[16]. Nonetheless, the precise mechanisms and molecular processes that modulate the AD and CV link remain elusive. A deeper understanding of their shared genetic architecture will provide insights into potentially common and distinct aetiologies of these conditions. Identifying shared targets and mechanisms that confer functional effects could lead to the discovery of interventions that address both neurodegenerative and CV diseases.

Here, we further investigated the commonalities in the genetic architecture of AD and CV traits and identified potential pleiotropic loci affecting multiple traits aiming to define common targets for therapeutic modulation. We explored a wide range of CV abnormalities and two common main risk factors, atherosclerosis and blood pressure (BP), proposed to underlie both CV and AD, to investigate different molecular pathways that may link different CV manifestations to AD. We performed a large-scale multi-trait GWAS analysis on AD and CV traits, followed by genetic colocalisation analysis to highlight candidate pleiotropic genes and their tissue sites of action. To further characterise biological pathways involved in both diseases, we leveraged data from single-cell RNA-sequence for differential gene expression with disease to explore relevant gene co-expression networks and protein-protein interactions in the brain and CV tissues. A schematic overview of the study is presented in Fig. 1.

## Results

### Multi-trait genetic association analysis identifies 5 novel AD loci and 9 shared loci between AD and CV traits

We performed five pairwise multi-trait analyses of GWAS (MTAG)[17] on AD and CAD, AF, stroke, carotid intima-media thickness (cIMT), and systolic and diastolic blood pressure (SBP, DBP). We examined the bivariate genetic correlation between AD and the examined CV traits (Supplementary Table 1) and visually illustrated the MTAG results alongside those from the original GWAS (Supplementary Fig. 1-6). Across all pairwise MTAG analyses, we identified 27 unique genetic loci associated with AD at genome-wide significance (GWS) level ($P < 5 \times 10^{-8}$) corresponding to 114 unique single-nucleotide polymorphisms (SNPs) (Supplementary Data 1). Out of the 27 AD loci, 5 were novel (not within ±500 kilobases (kb) of the previously known AD loci) and among them rs73069394 (*ULK4*) displayed the strongest association (Table 1).

To further validate the associations of the novel AD loci, we applied summary data-based Mendelian randomisation (SMR)[18] and heterogeneity in dependent instruments (HEIDI) using gene expression data from relevant brain tissues. SMR analysis suggested a potentially causal association between *ULK4* expression in the hippocampus and AD risk ($\beta_{SMR} = 0.04$, $P_{SMR} = 3.4 \times 10^{-10}$, $P_{HEIDI} = 0.22$). SMR was not possible for the remaining loci due to unavailability of gene expression data.

Furthermore, we found 1222 top signals associated with different CV traits at GWS level in 740 genetic loci (Supplementary Data 2). Of these, 13 novel loci were highlighted for CAD (*N* = 4), cIMT (*N* = 8) and stroke (*N* = 1) (Table 1). Overall, 15 of the unique AD SNPs (9 loci) were additionally associated at GWS level with at least one of the examined CV traits (Supplementary Data 3).

### A colocalisation analysis defines genetic loci shared by AD and different CV traits

Using the Hypothesis Prioritisation in multi-trait Colocalization (HyPrColoc)[19] method on all 767 MTAG-reported loci (27 AD + 740 CV), we identified 21 loci which colocalised between AD and CV traits with a posterior probability (PP) > 0.5 (Fig. 2, Supplementary Data 4). Most colocalised loci were found either between AD and AF (7 loci) or between AD and DBP (7 loci). Substantial evidence for colocalisation was observed for a locus at chr8:124,608,614 ± 200 kb (*RN7SKP155*) associated with AD and cIMT (PP = 1) and a locus at chr11:47,391,948 ± 200 kb (*SPI1*) associated with AD and DBP

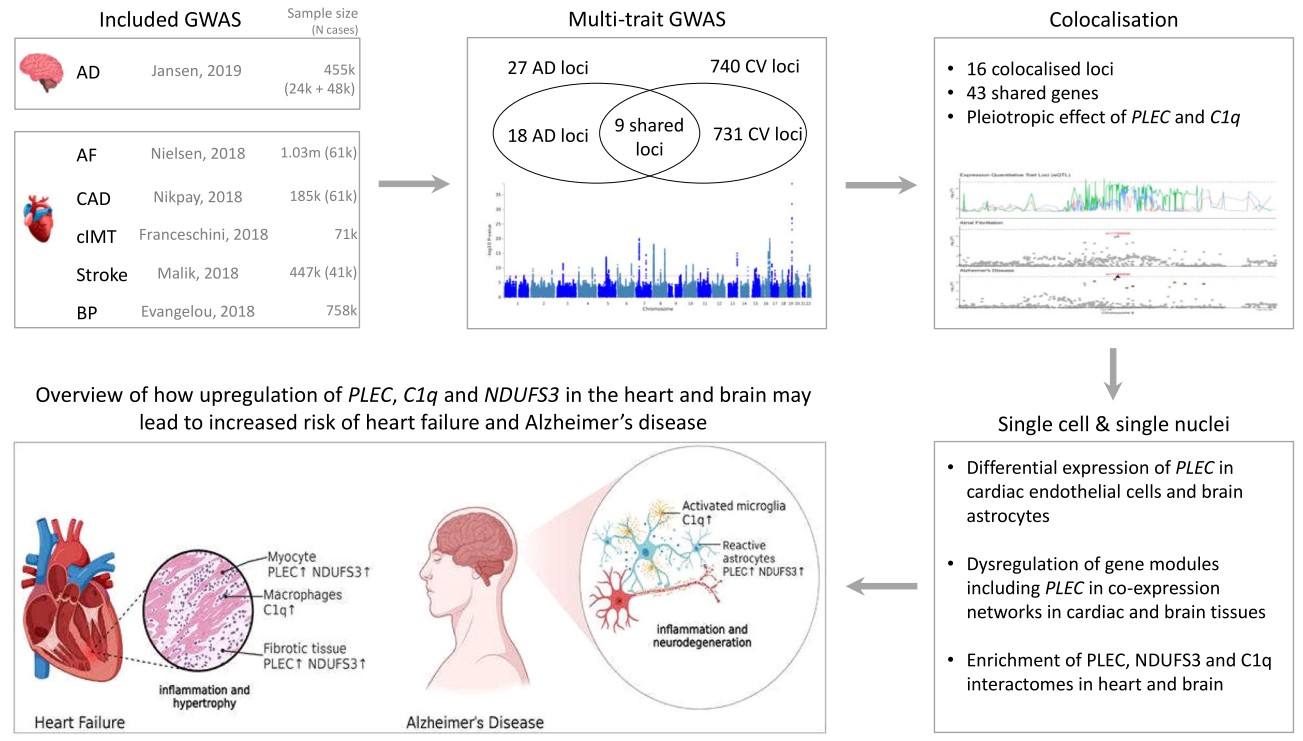

**Fig. 1 | Study design schematic overview.** AD Alzheimer's disease, AF atrial fibrillation, CAD coronary artery disease, cIMT carotid intima media thickness, BP blood pressure, CV cardiovascular, HF heart failure. Created in BioRender. Wang, D. (2023) BioRender.com/q29l509.

**Table 1 | Novel genetic loci associated with the examined traits in a genome-wide significance level (two-sided $P < 5 \times 10^{-8}$)**

| SNP | Chr | Pos | EA | OA | MAF | Beta | P | Gene |
|---|---|---|---|---|---|---|---|---|
| **Alzheimer's disease** | | | | | | | | |
| rs7529220 | 1 | 22282619 | T | C | 0.139 | −0.01 | $1.7 \times 10^{-8}$ | HSPG2 |
| rs11692604 | 2 | 19947507 | C | T | 0.483 | 0.01 | $4.1 \times 10^{-8}$ | AC019055.1 |
| rs73069394 | 3 | 41787233 | A | G | 0.188 | 0.03 | $1.5 \times 10^{-29}$ | ULK4 |
| rs77399788 | 5 | 123003001 | G | A | 0.058 | 0.03 | $1.8 \times 10^{-10}$ | KRT18P16 |
| rs56365761 | 19 | 39148103 | G | A | 0.436 | −0.01 | $4.6 \times 10^{-8}$ | ACTN4 |
| **Coronary artery disease** | | | | | | | | |
| rs2552527 | 2 | 218688596 | G | T | 0.409 | −0.02 | $2.0 \times 10^{-8}$ | TNS1 |
| rs748431 | 3 | 14928077 | G | T | 0.395 | 0.02 | $3.5 \times 10^{-8}$ | FGD5 |
| rs11723436 | 4 | 120901336 | G | A | 0.327 | 0.02 | $2.7 \times 10^{-8}$ | RP11-7OON1.1 |
| rs15052 | 19 | 41813375 | C | T | 0.158 | 0.02 | $3.6 \times 10^{-8}$ | HNRNPUL1:TGFB1 |
| **Carotid intima-media thickness** | | | | | | | | |
| rs10064683 | 5 | 95567760 | A | G | 0.352 | 0.02 | $1.3 \times 10^{-8}$ | CTD-2337A12.1 |
| rs6904596 | 6 | 27491299 | A | G | 0.082 | 0.04 | $1.2 \times 10^{-8}$ | HNRNPA1P1 |
| rs56118607 | 9 | 127898024 | A | G | 0.124 | 0.03 | $3.8 \times 10^{-8}$ | SCAI |
| rs1887182 | 10 | 97013497 | G | T | 0.467 | 0.02 | $1.0 \times 10^{-8}$ | PDLIM1 |
| rs11029956 | 11 | 27355804 | A | G | 0.340 | 0.02 | $7.4 \times 10^{-9}$ | CCDC34 |
| rs12370774 | 12 | 106510413 | T | C | 0.097 | −0.04 | $3.4 \times 10^{-8}$ | NUAK1 |
| rs76064118 | 19 | 2235284 | T | C | 0.053 | 0.06 | $4.8 \times 10^{-8}$ | PLEKHJ1 |
| rs1034565 | 22 | 19984211 | T | C | 0.284 | 0.03 | $9.8 \times 10^{-9}$ | ARVCF |
| **Stroke** | | | | | | | | |
| rs2284665 | 10 | 124226630 | T | G | 0.197 | −0.01 | $4.4 \times 10^{-8}$ | HTRA1 |

*SNP* Single-nucleotide polymorphism; *Chr* Chromosome; *Pos* Position; *EA* Effect allele; *OA* Other allele; *MAF* Minor allele frequency; *Beta* Effect size estimate; *P* Two-sided *P*-value; *Gene* Mapped gene.

(PP = 0.95). Among the 21 loci with evidence for colocalisation, there were three loci for which a single candidate causal variant explained a large proportion of the association: rs11786896 (mapped in *PLEC*; colocalised with AD-AF; PP = 0.97; 86% of PP explained by SNP), rs7529220 (mapped in *HSPG2*; colocalised with AD-AF; PP = 1; 90% of PP explained by SNP) and rs429358 (mapped in *APOE*; colocalised with AD-CAD; PP = 0.57; 93% of PP explained by SNP). Although rs11786896 in *PLEC* is not classified as a novel AD locus due to its proximity to a previously reported variant (rs34173062 in *SHARPIN*), the two loci are in linkage equilibrium ($r^2 = 0.006$) and the regional plots suggest it likely represents a different independent signal (Supplementary Fig. 7).

**Gene expression colocalisation analysis prioritises causal genes shared by AD and CV traits**

To identify potential pleiotropic causal genes for the colocalised loci, we tested the colocalisation of AD and CV traits with the expression of nearby genes in 48 tissues using expression quantitative trait loci (eQTL) data from Genotype-Tissue Expression (GTEx) in the 21 colocalised loci. We found that AD and at least one CV trait colocalised in 16 loci with expression of one or more genes in the same tissue, for a total of 53 associations with 43 genes (Supplementary Fig. 8, Supplementary Data 5). Of these, 20 associations were found for AF (in 6 loci with 17 genes), 22 for DBP (6 loci with 16 genes), 3 for stroke (1 locus with 3 genes) and 8 for cIMT (3 loci with 6 genes).

In two loci, a single candidate causal variant explained the colocalisation of AD, CV trait and tissue-specific gene expression: rs11786896 (*PLEC*) and rs7529220 (*HSPG2*). The intronic variant rs11786896 (*PLEC*) explained the colocalisation of AD and AF with expression levels of *PLEC* in the cardiac left ventricle (PP = 0.99, %PP explained by SNP = 99%) and skeletal muscle (PP = 0.92, %PP explained by SNP = 98%) (Fig. 3). rs11786896 was associated with increased risk of AD (Odds Ratio, OR = 1.02, $P = 5 \times 10^{-8}$), increased risk of AF (OR = 1.02, $P = 1.1 \times 10^{-6}$) and lower expression of *PLEC* in cardiac left ventricle

(Beta = −0.71, $P = 5.9 \times 10^{-13}$), as well as in skeletal muscle (Beta = −0.3, $P = 7.7 \times 10^{-7}$).

The intergenic variant rs7529220 (*HSPG2*) explained the colocalisation of AD and AF with expression levels of *C1QA* (PP = 0.85, %PP = 82%), *C1QB* (PP = 0.83, %PP = 97%) and *C1QC* (PP = 0.61, %PP = 99%) in breast mammary tissue. The variant was associated with higher risk of AD (OR = 1.01, $P = 1.7 \times 10^{-8}$), higher risk of AF (OR = 1.01, $P = 2.7 \times 10^{-10}$) and increased expression of *C1QA* (Beta = 0.19, $P = 2.4 \times 10^{-4}$), *C1QB* (Beta = 0.17, $P = 2.6 \times 10^{-4}$), and *C1QC* (Beta = 0.15, $P = 8.1 \times 10^{-4}$) in mammary tissue.

**PLEC and C1Q are differentially expressed in the left ventricle with heart failure**

Defining the cells in which target genes are expressed and directions of expression associated with disease risk and with disease is important for predicting the directions of effect for potential therapeutic modulation. Therefore, we tested the expression patterns of the 43 genes indicated by the colocalisation analysis across cell types in single-cell RNA from the heart to discover whether differences in differential expression with disease were consistent with those predicted for disease risk. *PLEC* was expressed in all cell types found in the cardiac left ventricle, while *C1Q* was expressed only in macrophages (Fig. 4A, B). *PLEC* was differentially expressed with heart failure (HF) relative to healthy controls with upregulated expression in the endothelium ($\log_2$ fold change, $\log_2$FC = 0.40, $P = 0.015$) but downregulated in macrophages ($\log_2$FC = −0.59, $P = 6.7 \times 10^{-5}$). There was only a trend for differential expression with HF in cardiomyocytes ($\log_2$FC = 0.91, $P = 0.06$). All *C1Q* associated genes were downregulated in cardiac macrophages (*C1QA*, $\log_2$FC = −1.39, $P = 1 \times 10^{-6}$; *C1QB*, $\log_2$FC = −1.34, $P = 3.5 \times 10^{-5}$; *C1QC*, $\log_2$FC = −1.28, $P = 1.4 \times 10^{-4}$).

We explored differential expression with HF further by constructing high dimensional weighted gene co-expression networks (WGCN) for *PLEC* and *C1Q* to identify modules of highly correlated

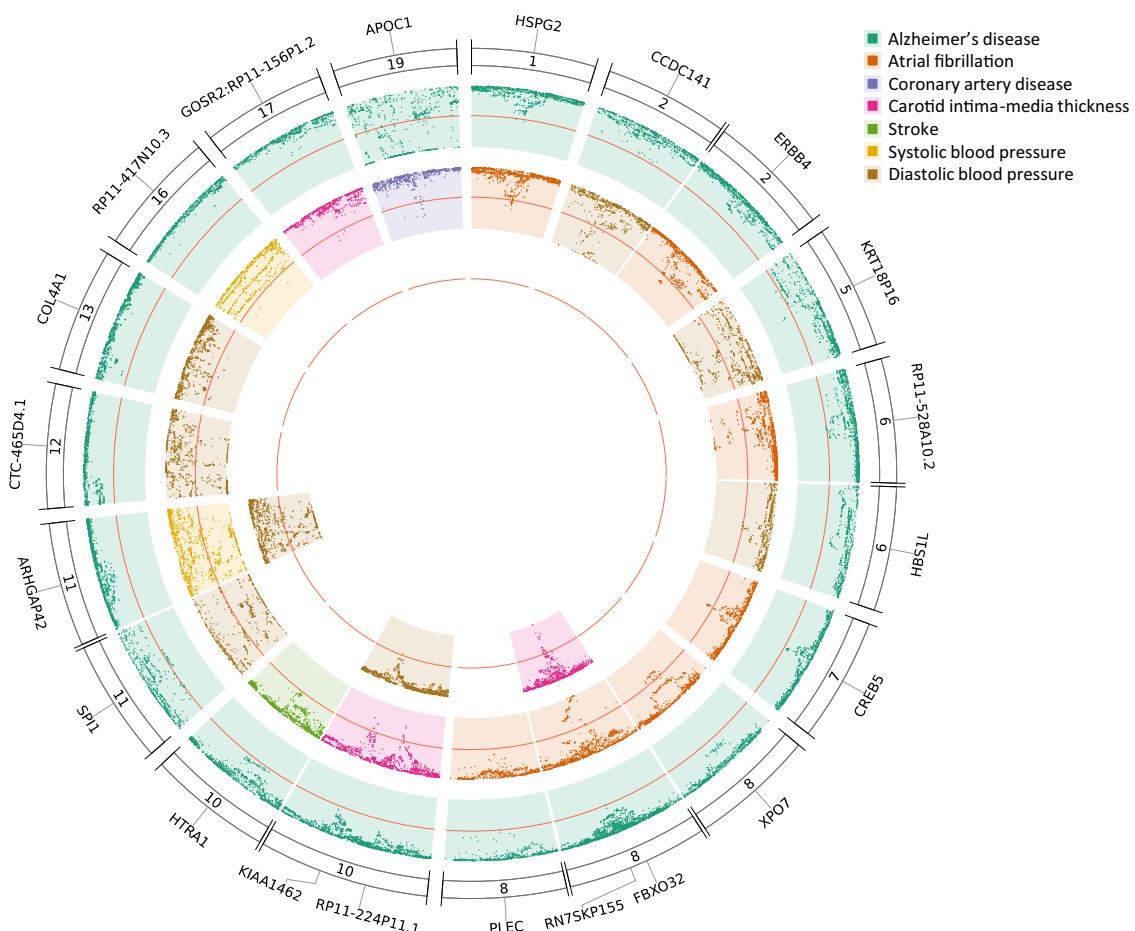

**Fig. 2 | Circular figure visualising regional plots on the colocalised loci between Alzheimer's disease (AD) and cardiovascular traits (CV).** The figure presents the distribution of *P*-values ($-\log_{10}P$) from MTAG with inner orientation (*P*-values are derived from two-sided statistical tests). The annotations show the mapped genes of the AD/CV top lead SNPs on the colocalised loci.

genes across cell types. We generated 12 gene co-expression modules in cardiac endothelial cells, 14 in cardiomyocytes and 6 in macrophages (Supplementary Data 6). A differential module eigengene analysis indicated that the module including *PLEC* was upregulated in cardiac vascular endothelial cells and cardiomyocytes in HF cases relative to the healthy controls and downregulated in macrophages with dilated cardiomyopathy (dCM) cases relative to healthy controls (Supplementary Data 7). The *C1Q*-containing module also was downregulated in cardiomyocytes in HF cases relative to healthy controls and downregulated in macrophages in dCM cases relative to healthy controls. Gene-set enrichments for biological processes defined the pathways most highly enriched in *PLEC*-containing modules (Supplementary Data 8), which included the "vascular endothelial growth factor receptor-2 signalling" and "endothelium development" pathways in endothelial cells (Fig. 4C) and many pathways related to mitochondrial oxidative metabolism in cardiomyocytes (Supplementary Fig. 9). In macrophages, the module including *C1Q* genes was enriched for "complement activation" and "synapse pruning" pathways, among others (Supplementary Fig. 10, Supplementary Data 9).

## PLEC and C1Q interactomes are enriched in cardiomyocytes, cardiac vascular endothelial cells and macrophages
To gain insights into the potential functional roles of proteins, we performed cell-specific protein-protein interaction (PPI) analyses on the set of colocalised candidate genes by constructing their protein interactomes across cell types of human cardiovascular tissue (Supplementary Data 10 and 11). The PLEC interactome was upregulated in

a pathway related to "ribosomal small subunit assembly" in endothelial cells (Fig. 4D) and upregulated in a "SRP-dependent co-translational protein targeting to membrane" pathway in cardiomyocytes (Supplementary Fig. 11). Additionally, an interactome containing both PLEC and NDUFS3 was enriched with HF and was upregulated in the pathways related to "aerobic electron transport chain" in endothelial cells (Fig. 4D) and "acetyl-CoA biosynthetic process from pyruvate" and "energy coupled proton transport" in cardiomyocytes (Supplementary Fig. 11). In macrophages, the PLEC-NDUFS3 interactome in HF was enriched for "mitochondria electron transport of cytochrome c to oxygen" while C1Q interactome was enriched for "cell junction disassembly" (Supplementary Fig. 12).

## PLEC is differentially expressed in brain astrocytes and is upregulated in AD
We also tested the expression of the 43 genes from our colocalisation analysis across different cell types in post-mortem human brain samples. *PLEC* was highly expressed in astrocytes and (to a lesser degree) in neurons. *C1Q* genes were expressed primarily in microglia (Fig. 5A, B). *PLEC* was significantly upregulated in astrocytes from AD donors relative to non-diseased control donors ($\log_2$FC = 1.01, $P$ = 0.003). *C1Q* genes were not significantly differentially expressed in microglia but consistently showed lower mean expression with AD (*C1QA*: $\log_2$FC = −0.46, $P$ = 0.23; *C1QB*: $\log_2$FC = −0.5, $P$ = 0.3; *C1QC*: $\log_2$FC = 0.14, $P$ = 0.9).

We explored the cell-specific differential expression of these and co-expressed genes further with WGCNA, which identified 14 gene

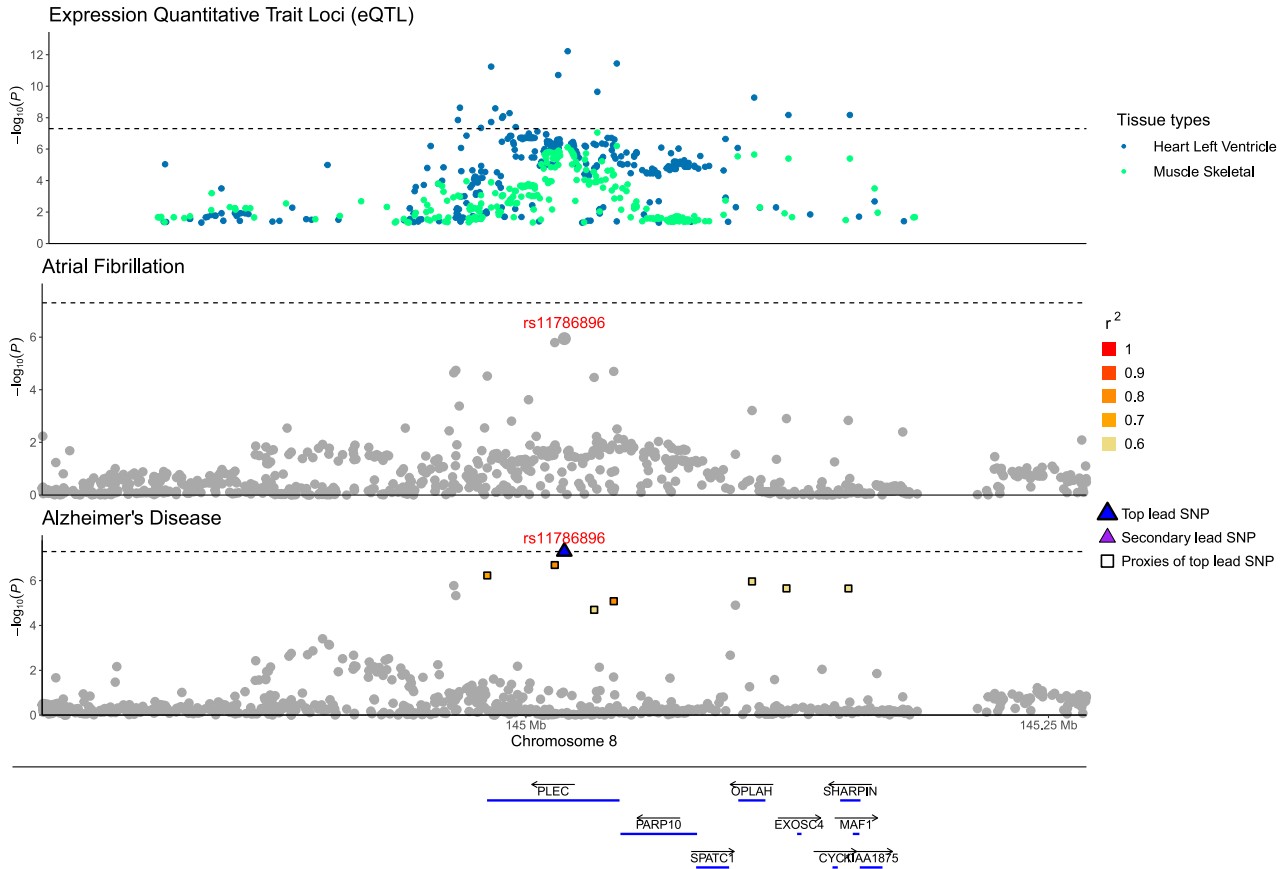

**Fig. 3 | Regional plot on the colocalised locus of candidate causal variant rs11786896 (mapped in *PLEC*) for Alzheimer's disease (bottom), atrial fibrillation (middle) and the expression quantitative trait loci (top) for the associated tissues.** The *x*-axis presents the chromosomal positions of single-nucleotide polymorphisms, while the *y*-axis shows the negative logarithm to base 10 of *P*-values ($-\log_{10}P$) from two-sided statistical tests.

co-expression modules in astrocytes and 8 in microglia (Supplementary Data 12). Differential module eigengene analyses showed that the *PLEC*-containing module was upregulated in astrocytes and the *C1Q*-containing module was downregulated in microglia (Supplementary Data 13). Gene-set enrichment for biological processes identified 25 associated pathways for the *PLEC*-containing module in astrocytes including "neuron projection morphogenesis" and "extracellular structure organisation" and 25 associated pathways for the *C1Q* gene module in microglia including the "positive regulation of intrinsic apoptotic signalling" (Supplementary Fig. 13, Supplementary Data 14).

We further explored the discordant directions in *C1Q* expression between eQTLs and post-mortem brain single nuclei data in relation to the beta-amyloid pathology load in microglia. For subjects with lower beta-amyloid load in the brain, we observed an increasing expression of *C1Q* in concordance with eQTL but subjects with higher beta-amyloid load showed decreased expression (Supplementary Fig. 14).

### PLEC, NDUFS3 and C1Q interactomes are enriched in astrocytes and microglia

A PPI analysis on candidate gene networks highlighted that the PLEC-NDUFS3 interactome also was enriched in both astrocytes and microglia in AD cases compared to controls (Supplementary Data 10, 15 and 16). Associated functional pathways were enriched for the "aerobic electron transport chain" and "SRP−dependent co-translational protein targeting to membrane" pathways in the astrocytes (Fig. 5D). In microglia, the C1Q interactome was enriched in "SRP−dependent co-translational protein targeting to membrane" pathway (Supplementary Fig. 15).

## Discussion

We adopted a comprehensive approach to understand the co-occurrence between AD and various CV diseases and traits based on several multi-trait GWAS to characterise their shared genetic architecture. Convergent evidence from colocalisation between AD, AF and eQTLs prioritised two genetic regions that each included a single candidate causal variant (rs11786896 which was an eQTL for *PLEC* and rs7529220 which was an eQTL for *C1QA*, *C1QB*, and *C1QC*) shared between AD and AF. Single-cell RNA-sequence data, co-expression network and protein-protein interaction analyses together were consistent in showing that *PLEC* is upregulated in left ventricular endothelium and cardiomyocytes with HF and in brain astrocytes with AD. By contrast, while *C1Q* genes are predicted to be upregulated with greater disease risk in cardiac macrophages for HF and in brain microglia for AD, we found opposite directions of difference with disease for both. We explored differences in the direction of changes from early disease (low beta-amyloid pathology load) to late (high beta-amyloid pathology load) for microglia and found the congruence with directions predicted for disease risk in early disease that was lost with in later disease progression. Our findings provide new insights into genetic pleiotropic effects and potential shared mechanisms causally related to both AD and CV traits.

Our study highlighted 5 not previously reported AD loci including SNPs located in or near *HSPG2*, *AC019055.1*, *ULK4*, *KRT18P16* and *ACTN4*. Of them, the intronic variant rs73069394 (*ULK4*) showed the strongest association with AD and was also GWS associated with DBP. The locus did not show evidence for colocalisation between AD and DBP suggesting this variant is not likely the causal one for both traits. Apart from DBP, GWAS studies have shown associations between this

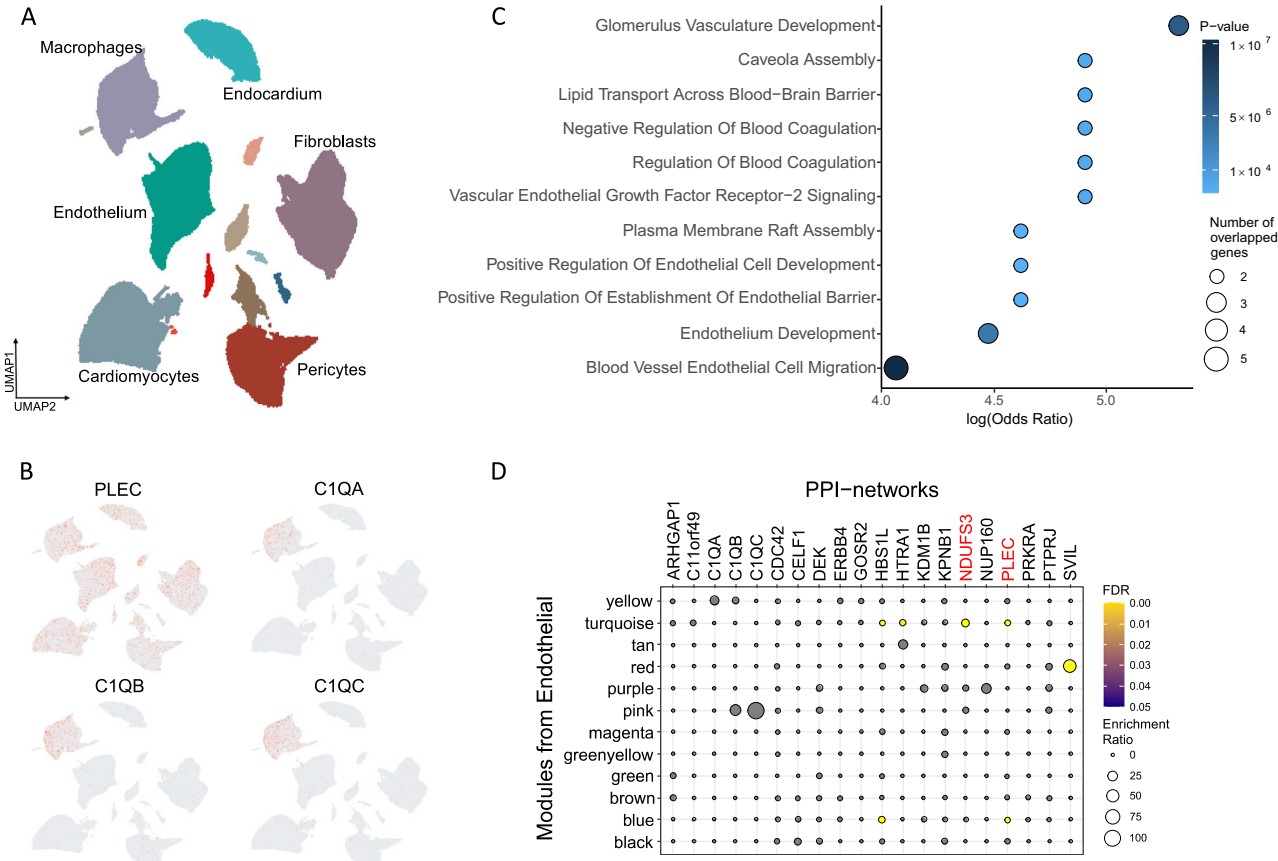

**Fig. 4 | Single-nuclei transcriptomes of cardiac tissue for genes increasing the risk of cardiomyopathy. A** The transcriptomes form discrete cell-specific clusters using Uniform Manifold Approximation and Projection (UMAP). **B** Expression of *PLEC* and *C1Q* genes across cell-specific clusters from UMAP. Red indicates higher expression. **C** Enriched pathways of the *PLEC*-containing module in cardiovascular endothelial cells between dilated cardiomyopathy cases and healthy controls (*P*-values are derived from two-sided statistical tests). **D** Enrichment of pathway genes in the protein interactomes of candidate genes in cardiovascular endothelial cells. The PLEC-interacting module is the brown and the C1q-interacting module is the pink.

locus and schizophrenia and bipolar disorder[20]. The role of the *ULK4* in relation to neurodegeneration is little studied but the gene function shows biological plausibility to AD. ULK4 protein is involved in the regulation of autophagy and plays multiple roles in brain function including neuronal growth, endocytosis and myelination which are pathways implicated to AD pathology[20,21]. Here, we also provided additional evidence from SMR analysis supporting a potentially causal association between expression levels of *ULK4* and AD.

Out of the several CV traits and diseases examined with AD, the largest number of pleiotropic signals with AD was observed for AF and BP highlighting the importance of pathways related to these CV traits in explaining comorbidities with AD. Numerous observational studies, provide growing evidence that BP and AF are associated with cognitive impairment, risk of AD and other dementias[22,23]. The suggested mechanistic links between these traits and AD involve a combination of cerebrovascular damage, neuroinflammation, amyloid-beta accumulation, oxidative stress, and endothelial dysfunction[24,25]. However, it is unclear whether the diseases have a shared pathophysiology or whether the relationship arises as downstream consequences of BP and AF (e.g., stroke).

Here we provide evidence suggesting a shared genetic determinant that may contribute to the pathophysiology of AF and AD. The colocalised intronic variant rs11786896 within the plectin gene (*PLEC*) was associated with lower expression of *PLEC* in the cardiac left ventricle (and skeletal muscle) and increased the risk of both AD and AF. Furthermore, *PLEC* was upregulated in left ventricular endothelium and cardiomyocytes in HF cases and in brain astrocytes in AD cases. A low-frequency missense variant in *PLEC* has been previously

associated with atrial fibrillation in a whole-genome sequencing data[26] whereas another missense variant has been linked to structural brain connectivity[27]. The intronic variant highlighted in this analysis has been previously associated with right ventricular structure and function[28] but has not been identified in GWAS studies as an AD or AF signal. *PLEC* is a member of a protein family, named plakins, with a crucial structural role in the cytoskeleton influencing cell architecture and tissue integrity and a partially functional role in the assembly, positioning, and regulation of signalling complexes[29,30]. Previous studies of human tissues or preclinical models provide independent evidence for an association of plectin with diseases including AD and AF[26,31].

Plectin is highly expressed in the central nervous system, especially at the interfaces between glia and pial cells and between glia and endothelial cells and is thought to be important to blood-brain barrier and pial surface integrity[32]. Plectin deficiency in mice has been associated with diminished learning capabilities and reduced long-term memory compared to wild-type littermates[33]. Here we provide evidence that the risk of AD may be affected via functions of plectin in astrocytes[34]. Astrocytes play multiple roles, central to the pathology of AD, including metabolic support for neurons, modulation of brain microvascular function and, through activities associated with those of microglia, inflammatory responses[34,35]. We hypothesise that these functional roles are mediated in part by interactions of plectin with intermediate filaments (IFs), microtubules and actin filaments[34]. IFs are important structural components of the cytoskeleton with crucial roles in synaptic activity, neurogenesis and repair after brain injury[36]. Differences in expression of plectin modulate neuronal function and

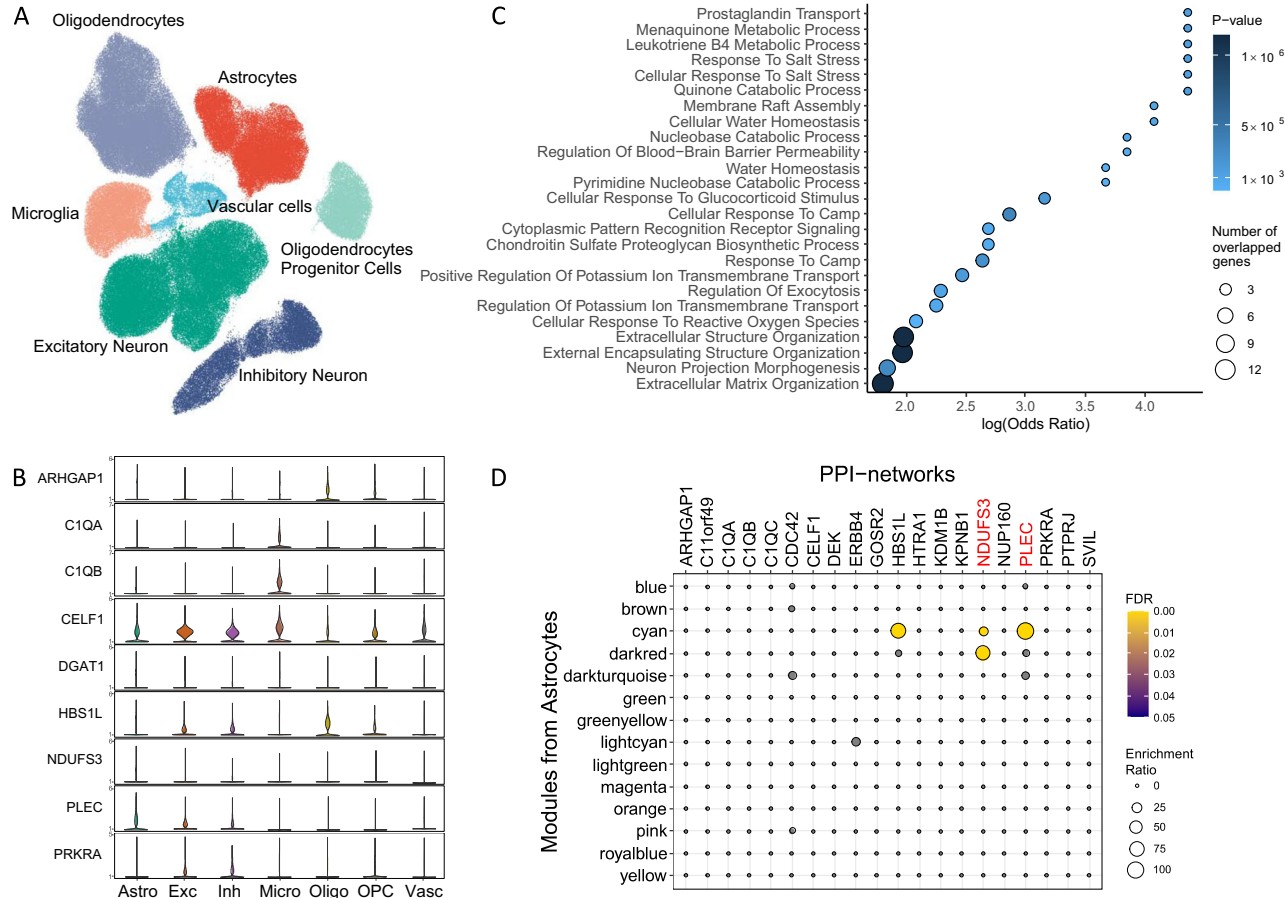

**Fig. 5 | Single-nuclei transcriptomes of human post-mortem brains for genes increasing the risk of Alzheimer's disease. A** The transcriptomes form discrete cell-specific clusters using Uniform Manifold Approximation and Projection (UMAP). **B** Expression of candidate genes across cell types in human brain. **C** Enriched pathways of the *PLEC*-containing module in astrocytes between Alzheimer's disease cases and healthy controls (*P*-values are derived from two-sided statistical tests). **D** Enrichment of pathway genes in the protein interactomes of candidate genes in astrocytes. The PLEC-interacting module is the cyan and the NDUFS3-interacting module is the dark red. Astro Astrocytes, Exc Excitatory Neuron, Inh Inhibitory Neuron, Micro Microglia, Oligo Oligodendrocytes, OPC Oligodendrocytes Progenitor Cells, Vasc Vascular cells.

vesicular trafficking generally and interactions with tau suggest potential roles specific to AD[33,37,38].

The role of *PLEC* in AF has been largely hypothesised to act via structural effects on the heart and cause electrophysiological abnormalities[26]. Here, we also show evidence for upregulation of *PLEC* in cardiomyocytes of HF patients. Therefore, in accordance with the hypothesised mechanisms linking *PLEC* to AD above, *PLEC* may play related roles in cardiomyocytes for assembling and mobilising the intermediate filaments and their networks. These effects further modulate contractile function in cardiomyocytes and inflammatory responses in macrophages which may further contribute to AF[39].

Another colocalised variant between AD and AF, the intergenic rs7529220, which is located 19 k upstream from Heparan Sulfate Proteoglycan 2 (*HSPG2*) and 21 k downstream from Chymotrypsin Like Elastase 3B (*CELA3B*), was associated with increased risk of AD and AF and higher expression of three genes of the Complement Component 1, Q Subcomponent (*C1Q*) family (*C1QA*, *C1QB*, *C1QC*) in breast mammary tissue (and, by inference, in brain vasculature) and is a previously unreported locus for AD. The variant is located 680 kb downstream of *C1Q* genes. The complement system plays a central role in synaptic remodelling in the brain and in cellular damage response more generally in the body[40,41]. We hypothesise that greater expression of *C1Q* may lead to higher activity of the complement system which in turn may potentiate synapse loss in early AD[42]. Similarly, C1Q has roles in the genesis of atherosclerotic plaques[43] and in the regulation of early

stages of inflammatory responses to the cardiomyocyte injury associated with a range of cardiac traits[44].

The variant rs429358, located within *APOE*, showed evidence for colocalisation between AD and CAD. However, further investigation using eQTL could not prioritise any shared gene within the locus expressed in the examined tissues. Given *APOE's* well-established role in AD and its potential involvement in other CV traits, including CAD[45], it is plausible that the locus affects the two traits independently through different pathways (horizontal pleiotropy) or through a shared gene expressed in tissues other than those examined in our study.

Our study had several strengths. First, we secured high statistical power for our study by including GWAS with substantial sample sizes ranging from 185,000 to 1,000,000 participants and we boosted the power even higher by performing suitable multivariate methods. Second, we combined advanced methods of genetic epidemiology and basic sciences and sought to provide supporting evidence from a variety of data. Third, exploring a wide spectrum of CV diseases pairwise enabled a comprehensive investigation into the diverse shared mechanisms underlying the relationship between different manifestations of CV conditions and AD.

However, a number of limitations also must be acknowledged. We restricted our analyses to a population of European ancestry. The lack of genetic diversity may have hampered the possibility of detecting other relevant variants. Additionally, we did not investigate a

considerable portion of the genetic predisposition coming from rare variants (MAF < 1%) as we excluded them from our analyses. This exclusion is a restriction of the MTAG method, in order to mitigate the risk of false-positive findings and biased results. Another limitation of using MTAG is that genetic variants that are not present in at least one GWAS dataset of each AD-CV pair were excluded from the corresponding pairwise MTAG analysis, and therefore some variants tested in one pairwise MTAG may not be tested in another. GWAS compares the cases we are studying with controls, but due to comorbidity, the number of other diseases might be higher in cases than in controls. As we used summary statistics without access to individual-level data, we couldn't examine or address this heterogeneity. Despite this limitation, the identified shared genes remain valid and important for understanding their genetic connections. Moreover, we used statistical methods to detect pleiotropy, and therefore considered a genetic locus pleiotropic if it was statistically significantly associated with two or more phenotypes. However, this approach for identification of pleiotropic genes may not always highlight shared biological pathways, as the identified genes could affect the traits independently via different pathways (horizontal pleiotropy), or they could even be expressed in different tissues in response to different signals[46,47]. The GWAS for AD combined clinically diagnosed cases of AD with AD-proxy status (inferred based on whether the parent was diagnosed with dementia). The correlation between the GWAS with and without the 'AD-by-proxy' cases was high[13] adding to the validity of this approach. Nonetheless, the inclusion of individuals with parental AD diagnoses in the original GWAS may have introduced greater heterogeneity and increased probability of misclassification, suggesting that some of our observed associations might be misestimated or influenced by other types of dementia. Considering the systemic nature of AD, which encompasses multiple pathways and various tissues, our study conducted a thorough investigation across all tissues, assuming correlations among many eQTLs across different tissues. However, it is important to acknowledge that some eQTLs from tissues not directly linked to AD or CV traits could represent false positive associations. To mitigate this, we validated our findings using single-cell and single-nuclei data obtained from brain tissue samples of AD cases and cardiac tissue samples from individuals with HF. Furthermore, due to a limited number of cells for specific cell types, we had to combine single-cell data from multiple samples. We focused on tissue samples that were already enriched for cardiomyocytes, endothelium, and macrophages. Moreover, the expression for some candidate genes in our data was limited and thus additional sequencing data and reads are needed to investigate them further. Additional RNA sequencing data of different AD and CV conditions would probably be even more informative. Finally, epidemiological data frequently encounter challenges in clarifying pathogenic mechanisms. Although we provided evidence from mechanistic experiments, the exact biological processes involved cannot be inferred. Further work is needed to validate our findings and the suggested disease pathways.

In conclusion, we performed a multi-trait analysis on AD and CV traits and a subsequent colocalisation analysis detecting 16 shared genetic loci and further prioritising two shared causal variants between the aforementioned traits. Our findings define shared mechanisms for AD and different CV diseases. The complement system has been explored as a target for preventive or disease-modifying therapies in CV disease[48] and AD[49]. Our work suggests that plectin or members of its interactome could offer new and potentially promising targets for preventive and therapeutic medicines with benefits across these common comorbid disorders.

## Methods
### Study population
We restricted our study to a population of European ancestry. We used the summary statistics from seven GWAS on the following diseases:

AD[13], AF[50], CAD[51], cIMT (see Supplementary Methods), stroke[52], SBP[53], and DBP[53]. We selected these datasets based on the sample size (with preference for larger GWAS), the quality of the phenotypic data, the date of publication (favouring more recent studies), and their relevance to our research questions. Supplementary Table 2 lists basic characteristics of all included GWAS datasets.

### Genotypic quality control
The seven initial datasets contained genotyped and imputed SNPs ranging from 7 to 34 million SNPs. We included in the analysis only SNPs that were present in both datasets (AD and the examined CV trait). Furthermore, we excluded all insertions, deletions, and rare variants (minor allele frequency; MAF < 0.01), variants with sample sizes < 2/3 of the 90th percentile and palindromic SNPs. Finally, more than 5.75 million SNPs were included in the analysis.

### Multi-trait association analysis
We performed five bivariate analyses on AD and a different each time CV trait (1. AF, 2. CAD, 3. cIMT, 4. Stroke, 5. SBP-DBP) using MTAG[17]. We calculated the genetic correlation between the traits and further corrected our data for sample overlap using bivariate linkage disequilibrium (LD) score regression as implemented in MTAG. Each MTAG analysis generated distinct trait-specific datasets (11 in total: 5 with AD plus 6 with CV traits) containing the trait-specific effect estimates for the included SNPs after leveraging for genetic correlation of the examined traits. As a result, the summary statistics from MTAG can be interpreted as similar to those from a univariate single-trait GWAS[17].

### Functional mapping and annotation
We used Functional Mapping and Annotation of GWAS (FUMA)[54] to functionally analyse all the generated summary results from MTAG. All the genome-wide significant (GWS) SNPs ($P < 5 \times 10^{-8}$) were initially clumped ($r^2 < 0.6$) to determine the coordinates of the genomic risk loci and then clumped again ($r^2 < 0.1$) to define independent signals. SNPs in pairwise LD at $0.1 \le r^2 < 0.6$ or SNPs located closer than 500 kb were assigned to the same LD block. SNPs that survived the second clumping were the independent signals. Independent SNPs with the smallest $P$-value in each LD block were defined as the top signals while the remaining were secondary signals. We further performed annotation and gene prioritisation analysis including all SNPs that survived the first clumping. We used the European sample of 1000 Genome Project Phase 3[55] to calculate pairwise LD between SNPs. SNPs were positionally mapped to their nearest protein-coding genes (Ensembl build v92).

To ensure the robustness of our findings, we considered only MTAG loci if the respective top variants were also associated with the examined trait ($P < 0.01$) in the corresponding original GWAS with a concordant direction of effect (between MTAG and the original GWAS). To identify the unique AD top and secondary independent signals for AD, we gathered independent signals from all pairwise MTAG analyses and excluded duplicate signals or proxies (within ±500 kb or LD $r^2 > 0.1$), keeping the strongest signal with the smallest $P$-value.

### Novel loci definition and replication
A signal indicated by MTAG was considered novel if its top variant achieved genome-wide significance ($P < 5 \times 10^{-8}$) in the MTAG results and was also significant ($P < 0.01$) in the included original univariate GWAS with a concordant direction of effect (between MTAG and the original GWAS). Additionally, the top variant should not be located within ±500 kb or in LD ($r^2 > 0.1$) with previously reported loci. For AD, in addition to the included GWAS study[13], novel loci were compared to two previously published key GWAS studies[56,57].

We applied SMR[18] and HEIDI analysis to investigate whether expressions of the novel AD MTAG-identified genes in relevant brain

tissues were causally associated with AD risk. SMR was performed by integrating summary eQTL data from GTEx version 8[58] in hippocampus and cortex (SNPs within 1 Mb of the transcription start site with $P < 1 \times 10^{-5}$) and AD summary statistics from MTAG.

### Trait-trait and trait-eQTL colocalisation analysis

We used HyPrColoc R package[19] to perform colocalisation analysis. HyPrColoc is a Bayesian divisive clustering algorithm for identifying shared genetic associations between traits in a genomic region using GWAS summary statistics. We performed this method to identify colocalised loci between AD and CV traits and prioritise causal variants explaining the shared association.

We performed a trait-trait colocalisation analysis for each top signal indicated from MTAG in a region ±200 kb from the top SNP. We considered variant-specific priors for our analyses, which assumes that the probability of a variant being colocalised with a set of traits decreases as the number of the set of traits increases. The variant-specific priors model requires the specification of two priors. We specified the prior probability that a variant is associated with a single trait only at $P = 1 \times 10^{-4}$ and a conditional prior probability that a variant is associated with an additional trait given that it is already associated with another trait at $P_c = 0.02$.

A PP > 0.5 was considered adequate evidence that the examined traits colocalise in the locus. We divided the evidence of colocalisation into two categories: (1) considerable evidence (0.5 < PP < 0.75) and (2) strong evidence (PP ≥ 0.75). To deal with spurious pleiotropy, we restricted the analyses to regions with at least one SNP with $P < 5 \times 10^{-4}$ in the respective univariate GWAS. Additionally, we visually inspected the colocalised loci by constructing suitable regional plots. The variants explaining at least 80% of the shared association were considered candidate causal. To limit the probability of false positive findings, we considered as causal the variants that were associated ($P < 0.01$) with both AD and the respective CV trait in the respective included univariate GWAS.

For the loci found to colocalise in the trait-trait colocalisation analysis, we further performed trait-expression quantitative trait loci (eQTL) colocalisation using AD, CV trait and eQTL from 48 tissues, retrieved from Genotype-Tissue Expression version 7 (GTEx v7), implementing the same parameters and approach as described in trait-trait colocalisation. The trait-eQTL colocalisation analysis was conducted to detect shared genes between the traits and investigate the tissues they are expressed.

### Single-cell and single-nuclei data acquisition

We used Gene Expression Omnibus[59] (GEO) to retrieve data for single cells of the left ventricle from 6 heart failure (HF) cases and 7 healthy controls[60], left ventricular single nuclei from 13 dilated cardiomyopathy (dCM) cases and 25 healthy controls[61], and post-mortem brain single nuclei from 9 AD cases and 8 healthy controls[62]. To deal with the small sample size in cardiac single-cell data, individuals with either coronary heart failure or dCM were considered HF cases.

### Quality control of single-cell and single-nuclei data

To quality control (QC) the data we implemented the scFlow pipeline[63]. Samples with < 100 cells in cardiac data and < 200 cells in brain data were removed. For brain data, Ambient RNA profiles were performed using EmptyDrops[64]. We restricted the minimum number of expressive features to 300 for cardiac data and 100 for brain data. For cardiomyocyte-enriched samples, we set the minimum library size per cell to 1000 while keeping the default value for the rest cell types. Only genes with a minimum of 2 counts in at least 3 cells were included. Doublet cells and non-annotated genes were removed. After QC, there was adequate sample to analyse cardiomyocyte-enriched (2,989 cells from 2 HF cases and 5 controls) and endothelium-enriched single cells (2,269 cells from 4 HF cases and 4 controls), macrophage single nuclei (207,345 nuclei from 13 dHF and 25 controls), astrocyte (34060 nuclei

from 25 AD cases and 24 controls), and microglia single nuclei (15292 nuclei from 25 AD cases and 24 controls).

### Single-cell and single-nuclei data integration, clustering and cell-type annotation

Cells that successfully passed the QC were integrated across samples using the linked inference of genomic experimental relationships (LIGER) method[65] defining a parameter lambda = 5 and selecting a sample-specific optimum value for parameter K: cardiomyocyte-enriched single cells ($K = 25$), endothelium-enriched single cells ($K = 30$), macrophage single nuclei ($K = 40$), and brain single nuclei ($K = 20$). A dimensionality reduction was performed by implementing the uniform manifold approximation and projection (UMAP)[66] algorithm to generate two-dimensional embeddings of the LIGER integrated factors using the first 10 principal components (PCs) on heart single cells, the first 60 PCs on heart single nuclei and the first 30 PCs on brain single nuclei. We subsequently detected cell clusters of the UMAP embeddings implementing the Leiden community detection algorithm[67] using a sample-specific parameter k: cardiomyocyte-enriched ($k = 9$) and endothelium-enriched single cells ($k = 10$), heart single nuclei ($k = 45$), and brain single nuclei ($k = 50$). Following clustering, we used the Expression Weighted Celltype Enrichment (EWCE)[68] algorithm to perform a cell-type prediction on cell clusters using previously reported reference datasets for cardiac single cells[69], cardiac single nuclei[61] and brain single nuclei[70].

### Differential gene expression analysis

A differential gene expression (DGE) analysis was performed separately for each cell-specific sample on all candidate genes detected from trait-eQTL colocalisation analysis. We investigated the expression of the genes in the RNAseq data and included only genes expressed in at least one cell type of the examined tissues. We followed a generalised linear mixed model approach as implemented in MAST[71] after excluding genes expressed in < 10% of cells. Units for differential expression were defined as $\log_2$ fold change ($\log_2$FC) per unit change of the respective contrast. We considered as meaningfully differentially expressed genes those with a $\log_2$FC ≥ 0.25 and a nominal $P$-value < 0.05.

### Weighted gene co-expression network

We further constructed cell type-specific co-expression networks on selected candidate genes using high dimensional weighted gene co-expression network analysis (hdWGCNA)[72] R package. We applied the K-Nearest Neighbours algorithm to identify groups of similar cells by means of transcriptomics (metacells) and constructed a metacell gene expression matrix. We constructed the co-expression network using the lowest soft power threshold that has a Scale Free Topology Model Fit ≥ 0.8. Genes that were not grouped into any co-expression module were excluded ("grey" module). We also excluded modules with < 20 genes. We obtained the module eigengene values, which describe the expression patterns of entire co-expression modules, and performed a differential module eigengenes analysis applying a Mann-Whitney U test. To reduce false-positive findings due to multiple testing inflation, we implemented the Benjamini-Hochberg false discovery rate (FDR) method[73]. We also conducted a pathway enrichment analysis using Enrichr v.3.0 R package[74] and analysed only gene-sets with at least 20 genes.

### Protein-protein interaction analysis

We used the STRINGdb[75] R package to analyse the full protein-protein interaction network data from STRING v11 database[76]. We expanded the candidate set of genes from the trait-eQTL colocalisation analysis by incorporating genes with protein-protein interactions (experimental evidence ≥ 700) with the candidates. Using the previously constructed cell type-specific modules from hdWGCNA, we performed

an enrichment analysis per module on candidate genes between the module and reference set using Fisher's exact tests. The Benjamini-Hochberg FDR approach was used to correct for type I error inflation due to the multiple testing error.

### Reporting summary

Further information on research design is available in the Nature Portfolio Reporting Summary linked to this article.

## Data availability

The summary statistics from the GWAS included in this study are publicly available and can be retrieved from GWAS Catalog under the accession codes GCST007320 (AD), GCST006414 (AF), GCST003116 (CAD), GCST006906 (stroke), GCST006624 (SBP) and GCST006630 (DBP). Heart single-cell data from the left ventricle tissue were downloaded from Gene Expression Omnibus (GEO) under accession codes GSE109816 (cardiomyocytes-enriched samples) and GSE121893 (normal digested samples). Single-nuclei data for the left ventricular tissue were retrieved from GEO under accession code GSE109816. The single-nuclei RNA sequencing data for human post-mortem brain samples from AD and Control samples were retrieved from GEO under accession code GSE160936. The MTAG summary statistics generated in this study have been deposited in NHGRI-EBI GWAS Catalog under accession codes GCST90449053 (AD from AD-AF MTAG), GCST90449054 (AF), GCST90449055 (AD from AD-BP MTAG), GCST90449056 (SBP), GCST90449057 (DBP), GCST90449058 (AD from AD-CAD MTAG), GCST90449059 (CAD), GCST90449060 (AD from AD-cIMT MTAG), GCST90449061 (cIMT), GCST90449062 (AD from AD-stroke MTAG), and GCST90449063 (stroke). All other data generated in this study are provided with this published article (and its supplementary information files).

## Code availability

No previously unreported custom computer code or mathematical algorithm was used to generate results central to the conclusions.

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

## Acknowledgements

The Genotype-Tissue Expression version 7 (GTEx v7) Project was supported by the Common Fund of the Office of the Director of the National Institutes of Health, and by NCI, NHGRI, NHLBI, NIDA, NIMH, and NINDS. The data used for the analyses described in this manuscript were obtained from dbGaP accession number phs000424.v7.p2 on 30/03/2022. DW is supported by the Academy of Medical Sciences

Professorship (APR7_1002). PMM and PE acknowledge personal support from the UK Dementia Research Institute, which is funding by the UKRI Medical Research Council, Alzheimer's Society and Alzheimer's Research UK, Edmond J. Safra Foundation and Lily Safra, an NIHR Senior Investigator Award, and the Imperial College Healthcare Trust (ICHT) NIHR Biomedical Research Centre. The authors acknowledge support from the Trustees of the Sir Michael Uren Foundation for this work. The authors also acknowledge support by the British Heart Foundation Research Excellence Award (4) (RE/24/130023 and the NIHR Imperial Biomedical Research Centre (BRC).

## Author contributions

A.D., D.W., P.M., F.K. and I.T. designed the research. F.K., N.F., P.F.T., and D.M. conducted the analyses and visualised the results. F.K., N.F., P.F.T., E.E., A.D., D.W., P.M., and I.T. interpreted the results. F.K. wrote the manuscript. N.F., P.F.T., D.M., P.E., A.D., D.W., P.M., and I.T. critically revised the manuscript.

## Competing interests

The authors declare no comparing interests.
