## [Transparent Peer Review file · Nature Communications]

Multi-trait association analysis reveals shared genetic loci between Alzheimer's disease and cardiovascular traits

Corresponding Author: Dr Fotios Koskeridis

Version 0:

Reviewer comments:

Reviewer #1

(Remarks to the Author)

The manuscript's exploration of genetic risk factors common to Alzheimer's disease and cardiovascular diseases is of significant interest due to the potential implications for understanding disease mechanisms and informing therapeutic strategies. However, the manuscript primarily recounts experimental methodologies and fails to provide substantial interpretation or discussion of the results. Moreover, the proposed shared risk factors, PLEC and C1Q, lack sufficient validation within the study. As it stands, I cannot endorse this manuscript for publication in Nature Communications.

To improve the manuscript, the authors should consider the following:

First of all, the GWAS database used for the analysis should be defined and interpreted carefully in the manuscript.

1. The author used published GWAS data from Alzheimer's diseases for the extraction of AD traits (62 AD genetic loci). The author should compare their analysis and the result with the original publication.
2. To extract cardiovascular traits, the author used diverse GWAS data from five different publications; atrial fibrillation, coronary artery disease, stroke, carotid intima-media thickness, systolic and diastolic blood pressure. 'Intima-media thickness' and 'systolic and diastolic blood pressure' are not terms of disease condition. Their inclusion in cardiovascular trait extraction necessitates a clear exposition of their connection to atrial fibrillation, coronary artery disease, and stroke.
3. The justification for using a diverse range of GWAS data to extract cardiovascular traits must be elaborated upon. The authors should demonstrate the advantage of this diversity and whether it led to definitive conclusions about cardiovascular traits.
4. A discussion on the 740 cardiovascular loci identified should be provided to underscore their relevance and significance.
5. The manuscript must critically examine the heterogeneity of the phenotypes in the GWAS datasets, especially considering the comorbidity of AD and cardiovascular symptoms, and delineate how these datasets were selected and utilized in light of the study's objectives

Regarding the identification of PLEC and C1Q as shared risk factors:

6. The significance of PLEC, HSPG2, and APOE, which were implicated based on the multi-trait colocalization method, must be contextualized with previous GWAS findings on AD and cardiovascular disease (CVD) traits.
7. The shared risk factors for AD and CVD indicated by genetic variants rs11786896 (PLEC) and rs7529220 (HSPG2) should be reassessed, as the observed odds ratios do not robustly categorize them as high-risk factors for Alzheimer's disease or atrial fibrillation.
8. The relevance of the identified loci to other cardiovascular traits should be discussed. If no correlation exists, the study should narrow its focus to the relationship between AD and atrial fibrillation, avoiding unnecessary data and analyses. Finally, while the single-nuclei transcriptome and protein-protein interaction analyses involving PLEC, NDUFS3, and C1Q on AD brain tissues are noted, the study fails to elucidate a clear pathogenic connection of these targets to Alzheimer's disease and atrial fibrillation.

Reviewer #2

(Remarks to the Author)

Overview

The work uses MTAG to examine AD with cardiovascular traits. The motivation is the long-standing association between cardiovascular disease and AD, and the method harnesses the genetic correlation to improve power for detecting genetic associations. The authors use eQTLs identified from a range of tissues from the GTEx project plus their results to perform colocalization analyses and hone in on two sites, and follow up some of those findings with analysis of gene expression. Overall, the manuscript was clear and interesting. The methods are well-described and approach seems sound, but it looked like there might be a disagreement between figure 3 and text and figure 2 does not share enough detail so it cannot be used to check against the text, which were my most important concerns. I had some specific comments below but I also wondered 1) if the authors could examine genetic correlation between AD and the traits in addition to sharing the MTAG results; and, 2) whether the authors considered testing colocalization with multiple traits at loci that seemed like they had ≥ 2 signal with several cardiovascular traits.

Major Comments

1. Title – at best, these data let you say that there is evidence consistent with what you'd observe with pleiotropy. Also, there are more genes implicated than just PLEC and C1Q, unless I missed something.
2. Figure 1 would be helped by sharing the sample sizes for each GWAS used to let the reader understand their magnitudes.
3. Figure 2 is difficult to read - can it just be made into a horizontal plot with labeled axes?
4. Figure 3 seems to present different p-values compared to the text (lines 112-115) that should be checked.
5. Please provide manhattan and qq plots for each trait combination.
6. Discussion - lines 218-221. The authors claim the results of their study disentangles the relationship between AD and AF. This is beyond what I think they can say with these data since the outcome in the Jansen paper is mainly from the UKB that infers case status using a question about whether the parent had dementia. This introduces a substantial amount of heterogeneity as to the underlying cause of dementia that ranges from cerebrovascular disease to a purely neurodegenerative cause, which preclude the kind of conclusion the authors made. Using Kunkle et al., 2018, which relies on a clinical diagnosis, might help this. Regardless of the specific cause of dementia, it's clear that there's a relationship between dementia and AF.
7. Using gene expression from tissues that are not likely directly relevant for the pathogenesis of AD or one of the CVD traits rests on an assumption that the eQTLs in that tissue are generalizable. It could be that they are false positives in one tissue and are correctly null in other tissues.

Minor Comments

1. Consider adding the limitation that not all GWAS summary results are complete so signal that is not present in one of the dataset pairs would not be tested using MTAG.
2. Figure 3 – the eQTL line plot is confusing. Why not stack the two tissues and use dots? Either way, the association between the eQTL signal and other traits (i.e., AD, atrial fib) are not striking.
3. Unlike CVD traits, AD has one region, the APOE locus, that has an outsized effect compared to other genetic loci. I would guess the results from APOC1 were driven by the association between AD and APOE.
4. Discussion - lines 204-5 (grammar/phrasing) - "rs11786896 expressed via PLEC and rs7529220 expressed via C1QA, C1QB, and C1QC" sounds odd. I'd phrase it something like rs1179686 is an eQTL for PLEC.

Version 1:

Reviewer comments:

Reviewer #1

(Remarks to the Author)

The authors have fully addressed reviewers' comments and the revised version is worth being accepted for publication as it is.

Reviewer #2

(Remarks to the Author)

No matter how AD is commonly defined (e.g., clinically or pathologically), it often co-occurring with evidence of cerebrovascular disease yet whether this is pleiotropic or causal relationship is unknown -- although there is some work that suggests a direct link <https://pubmed.ncbi.nlm.nih.gov/35772923/>.

The current work focuses on finding shared genetic risk using MTAG, which uses genetic correlation to boost power for multi-trait gwas.

The most interesting portion of the work is the MTAG results along with the SMR analysis.

By contrast the differential expression and network-based analysis was descriptive without addressing the most interesting question of the work -- the shared risk for CV traits and AD dementia.

Overall, the revised manuscript nicely addressed my comments.

Minor Comments

1. line 297 - AD-proxy status was inferred based on whether the parent was diagnosed with dementia (not AD per se).

REVIEWER COMMENTS

Reviewer #1 (Remarks to the Author):

The manuscript's exploration of genetic risk factors common to Alzheimer's disease and cardiovascular diseases is of significant interest due to the potential implications for understanding disease mechanisms and informing therapeutic strategies. However, the manuscript primarily recounts experimental methodologies and fails to provide substantial interpretation or discussion of the results. Moreover, the proposed shared risk factors, PLEC and C1Q, lack sufficient validation within the study. As it stands, I cannot endorse this manuscript for publication in Nature Communications. To improve the manuscript, the authors should consider the following:

Reply: Thank you for your feedback and insightful comments. We have carefully revised our manuscript to address them, ensuring a cautious and robust interpretation of our results. Further details on your specific comments are provided below.

1.1 First of all, the GWAS database used for the analysis should be defined and interpreted carefully in the manuscript.

Reply1.1: We have provided detailed descriptions of the GWAS datasets in the methods section, including citations, sources, and characteristics (new Supplementary Table 18).

In lines 327-332 (p.14) we now explain *"We restricted our study to a population of European ancestry. We used the summary statistics from seven GWAS on the following diseases: AD, AF, CAD, cIMT, stroke, SBP, and DBP. We selected these datasets based on the sample size (with preference for larger GWAS), the quality of the phenotypic data, the date of publication (favouring more recent studies), and their relevance to our research questions. Supplementary Table 18 lists basic characteristics of all included GWAS datasets."*

1.2 The author used published GWAS data from Alzheimer's diseases for the extraction of AD traits (62 AD genetic loci). The author should compare their analysis and the result with the original publication.

Reply 1.2: The primary focus of our results was the investigation of pleiotropic loci between AD and CVD rather than the description of novel findings stemming from MTAG. We acknowledge that the latter is also of interest to the readers. In this revised version, we have included a comprehensive comparison between novel and previous findings for all examined traits. We provided Manhattan plots and QQ-plots for each trait, comparing MTAG results with those from the original GWAS (Supplementary Figures 1-6) and reported novel loci for each trait (Table 1). We also provided new analysis to support the validity of novel findings in relation to AD.

To ensure robust findings and to facilitate a rigorous comparison, we implemented strict criteria to define MTAG loci. Specifically, we considered only loci that were at least 500kb distant and independent ($LD < 0.1$) from previously reported signals. Next to the genome-wide significant p-value in MTAG results, these loci also had to exhibit nominal significant association ($P < 0.01$) in the original GWAS with concordant direction of effect. To reflect these updates, we have thoroughly revised our manuscript accordingly. We have included a detailed explanation in the methods. For example:

“To ensure the robustness of our findings, we considered only MTAG loci if the respective top variants were also associated with the examined trait ($P < 0.01$) in the corresponding original GWAS with a concordant direction of effect (between MTAG and the original GWAS).” (p.15, lines 361-363)

Moreover, to identify novel signals, we compared the MTAG findings with those from previously published GWAS. Specifically for AD, we assessed previously identified loci from three key GWAS studies (Jansen et al., Kunkle et al., and Bellenguez et al.). A dedicated section in the methods has been included to elaborate on how novel signals were defined.

“A signal indicated by MTAG was considered novel if its top variant achieved genome-wide significance ($P < 5 \times 10^{-8}$) in MTAG results and was also statistically significant ($P < 0.01$) in the included original univariate GWAS with a concordant direction of effect (between MTAG and the original GWAS). Additionally, the top variant should not be located within $\pm 500\text{kb}$ or in LD ($r^2 > 0.1$) with previously reported loci. For AD, in addition to the included GWAS study, novel loci were compared to two previously published key GWAS studies.” (p.16, lines 368-373)

In the results section we highlight novel findings, for example:

“Across all pairwise MTAG analyses, we identified 27 unique genetic loci associated with AD at genome-wide significance (GWS) level ($P < 5 \times 10^{-8}$) corresponding to 114 unique single-nucleotide polymorphisms (SNPs) (Supplementary Table 2). Out of the 27 AD loci, 5 were novel (not within ± 500 kilobases (kb) of the previously known AD loci) and among them rs73069394 (ULK4) displayed the strongest association (Table 1).

To further validate the associations of the novel AD loci, we applied summary data-based Mendelian randomisation (SMR) and heterogeneity in dependent instruments (HEIDI) using gene expression data in relevant brain tissues. SMR analysis suggested a potentially causal association between ULK4 expression in the hippocampus and AD risk ($\beta_{SMR} = 0.04$, $P_{SMR} = 3.4 \times 10^{-10}$, $P_{HEIDI} > 0.05$). SMR was not possible for the remaining loci due to unavailability of gene expression data.

Furthermore, we found 1,222 top signals associated with different CV traits at GWS level in 740 genetic loci (Supplementary Table 3). Of these, 13 novel loci were highlighted for CAD ($N = 4$), cIMT ($N = 8$) and stroke ($N = 1$) (Table 1). Overall, 15 of the unique AD SNPs (9 loci) were additionally associated at GWS level with at least one of the examined CV traits (Supplementary Table 4).” (p.2-3, lines 47-63)

Of note, these refinements, which ensure our manuscript presents findings with robust evidence, do not alter the highlighted results or conclusions of our initial manuscript, as we had already applied these criteria in the subsequent analyses following MTAG.

Finally, we discuss novel findings:

“Our study highlighted 5 not previously reported AD loci including SNPs located in or near HSPG2, AC019055.1, ULK4, KRT18P16 and ACTN4. Of them, the intronic variant rs73069394 (ULK4) showed the strongest association with AD and was also GWS associated with DBP. The locus did not show evidence for colocalization between AD and DBP suggesting this variant is not likely the causal one for both traits. Apart from DBP, GWAS studies have shown associations between this locus and schizophrenia and bipolar disorder. The role of the ULK4 in relation to neurodegeneration is little studied but the gene function shows biological plausibility to AD. ULK4 protein is involved in the regulation of autophagy and plays multiple roles in brain function including neuronal growth, endocytosis and myelination which are pathways implicated to AD pathology. Here, we also provided additional evidence from SMR analysis supporting a potentially causal association between expression levels of ULK4 and AD.” (p.9, lines 196-207)

1.3 To extract cardiovascular traits, the author used diverse GWAS data from five different publications; atrial fibrillation, coronary artery disease, stroke, carotid intima-media thickness, systolic and diastolic blood pressure. ‘Intima-media thickness’ and ‘systolic and diastolic blood pressure’ are not terms of disease condition. Their inclusion in cardiovascular trait extraction necessitates a clear exposition of their connection to atrial fibrillation, coronary artery disease, and stroke.

Reply 1.3: Our objective was to systematically explore the interconnection between AD and CVDs by covering a broad spectrum of cardiovascular conditions and associated traits, which is now stated more clearly throughout the manuscript:

“AD has been considered a brain-specific disease whose primary pathology is confined to the brain. However, accumulating evidence suggests mechanistic links between a wide range of cardiovascular (CV) abnormalities and AD. Epidemiological studies and experimental data have shown consistent associations between manifestations of clinical CV diseases such as coronary heart disease (CHD), atrial fibrillation (AF) and stroke, with higher risk of AD. Several hypotheses have been proposed to explain this. Indeed, atherosclerosis, the main underlying cause of cardiovascular diseases, also has profound consequences on the cerebrovascular system. These include reduced blood flow and potential vascular damage in the

brain, impaired cerebral perfusion, and associations with inflammation and oxidative stress, all of which are factors that can contribute to neurodegenerative pathology and increase the risk of AD. Beyond atherosclerosis, other hemodynamic effects associated with hypertension, arteriosclerosis and subsequent aortic stiffening have been associated with cerebrovascular damage and cognitive function, potentially accelerating the onset and progression of AD.” (p.1, lines 4-17)

“Here, we further investigated the commonalities in the genetic architecture of AD and CV traits and identified pleiotropic loci affecting multiple traits aiming to define common targets for therapeutic modulation. We explored a wide range of CV abnormalities and two common main risk factors, atherosclerosis and blood pressure, proposed to underlie both CV and AD, to investigate different molecular pathways that may link different CV manifestations to AD.” (p.2, lines 29-33)

1.4 The justification for using a diverse range of GWAS data to extract cardiovascular traits must be elaborated upon. The authors should demonstrate the advantage of this diversity and whether it led to definitive conclusions about cardiovascular traits.

Reply 1.4: We selected a diverse range of GWAS datasets to encompass a broad spectrum of cardiovascular traits, given the unclear connection with AD. This approach allowed a thorough exploration of how genetic factors contribute to the interconnection between AD and CV conditions, offering a more comprehensive understanding than focusing solely on a single CV trait.

In the introduction, we have added a paragraph explaining this:

“Here, we further investigated the commonalities in the genetic architecture of AD and CV traits and identified pleiotropic loci affecting multiple traits aiming to define common targets for therapeutic modulation. We explored a wide range of CV abnormalities and two common main risk factors, atherosclerosis and blood pressure, proposed to underlie both CV and AD, to investigate different molecular pathways that may link different CV manifestations to AD.” (p.2, lines 29-33)

In the discussion, we further mention:

“Third, exploring a wide spectrum of CV diseases pairwise enabled a comprehensive investigation into the diverse shared mechanisms underlying the relationship between different manifestations of CV conditions and AD.” (p.12, lines 275-277)

1.5 A discussion on the 740 cardiovascular loci identified should be provided to underscore their relevance and significance.

Reply 1.5: Thank you for the suggestion. The primary aim of our study was to identify pleiotropic genes linking AD and CVDs. The genetic architecture of CV conditions has been much more extensively studied with large scale GWAS compared to AD. Here, we aimed to leverage the co-occurrence of CV conditions with AD to increase the power of AD GWAS studies and identify shared loci including novel loci related to AD. Following your suggestion, we now compared the reported loci for all examined traits (see response to comment 1.2). The vast majority of the 740 loci (727 loci) were already reported loci for CVD and the new ones are highlighted in Table 1.

1.6 The manuscript must critically examine the heterogeneity of the phenotypes in the GWAS datasets, especially considering the comorbidity of AD and cardiovascular symptoms, and delineate how these datasets were selected and utilized in light of the study's objectives

Reply 1.6: Our study utilised summary statistics derived from publicly available GWAS datasets, which precluded us from examining the heterogeneity in the original GWAS datasets. We acknowledge that any potential heterogeneity present in these datasets could affect the results of our study. We included this limitation in our manuscript.

“GWAS compares the cases we are studying with controls, but due to comorbidity, the number of other diseases might be higher in cases than in controls. As we used summary statistics

without access to individual-level data, we couldn't examine or address this heterogeneity. Despite this limitation, the identified shared genes remain valid and important for understanding their genetic connections.” (p.12, lines 286-291)

To mitigate this limitation, we employed a bivariate analysis approach, examining the shared associations between AD and each CV trait separately. We used MTAG, which accounts for phenotypic heterogeneity from diverse populations and studies by leveraging the genetic correlation among the examined traits. Furthermore, MTAG provides trait-specific summary statistics and therefore we were able to explore the potential influence of other CV traits on our observed associations between AD and specific CV traits, by systematically comparing our findings across all examined CV traits. This approach provided insights into the complex interrelationships among these traits and their potential impact on AD. Our discussion now states:

“Out of the several CV traits and diseases examined with AD, the largest number of pleiotropic signals with AD was observed for AF and BP highlighting the importance of pathways related to these CV traits in explaining comorbidities with AD. Numerous observational studies, provide growing evidence that BP and AF are associated with cognitive impairment, risk of AD and other dementias. The suggested mechanistic links between these traits and AD involve a combination of cerebrovascular damage, neuroinflammation, amyloid-beta accumulation, oxidative stress, and endothelial dysfunction. However, it is unclear whether the diseases have a shared pathophysiology or whether the relationship arises as downstream consequences of BP and AF (e.g., stroke).” (p.9, lines 208-216)

Finally, in response to your suggestion, we have provided a description in the methods outlining the criteria used for selecting and utilising the GWAS datasets relevant to our study objectives.

“We selected these datasets based on the sample size (with preference for larger GWAS), the quality of the phenotypic data, the date of publication (favouring more recent studies), and their relevance to our research questions.” (p.14, lines 329-332)

Regarding the identification of PLEC and C1Q as shared risk factors:

1.7 The significance of PLEC, HSPG2, and APOE, which were implicated based on the multi-trait colocalization method, must be contextualized with previous GWAS findings on AD and cardiovascular disease (CVD) traits.

Reply 1.7: Thank you for your comment. PLEC and HSPG2 are among our most highlighted findings, and we have discussed their importance in the context of previous studies. For instance, we state in the discussion:

“A low-frequency missense variant in PLEC has been previously associated with atrial fibrillation in whole-genome sequencing data whereas another missense variant has been linked to structural brain connectivity. The intronic variant highlighted in this analysis has been previously associated with right ventricular structure and function but has not been identified in GWAS studies as an AD or AF signal.” (p.10, lines 222-226)

“Previous studies of human tissues or preclinical models provide independent evidence for an association of plectin with diseases including AD and AF.” (p.10, lines 229-231)

We also included in the discussion regarding HSPG2:

“Another colocalised variant between AD and AF, the intergenic rs7529220, which is located 19k upstream from Heparan Sulfate Proteoglycan 2 (HSPG2) ... and is a previously unreported locus for AD.” (p.11, lines 252-257)

APOE is a well-established gene associated with AD, and previous studies have supported its involvement in cardiovascular traits. Although our initial colocalisation analysis provided considerable evidence for its possible pleiotropic effect on AD and coronary artery disease (PP = 0.57), our subsequent eQTL colocalisation analysis could not replicate this association. This suggests that our study could not prioritise any shared gene within the locus between the two traits expressed in the examined tissues. Given the robust evidence from several independent studies linking APOE to both traits, it is plausible that the locus affects the two traits independently through different pathways (horizontal pleiotropy).

Alternatively, there may be a shared gene expressed in tissues other than those we examined in our study. We added in the discussion:

“The variant rs429358, located within APOE, showed evidence for colocalisation between AD and CAD. However, further investigation using eQTL could not prioritise any shared gene within the locus expressed in the examined tissues. Given APOE’s well-established role in AD and its potential involvement in other CV traits, including CAD, it is plausible that the locus affects the two traits independently through different pathways (horizontal pleiotropy) or through a shared gene expressed in tissues other than those examined in our study.” (p.11, lines 264-269)

1.8 The shared risk factors for AD and CVD indicated by genetic variants rs11786896 (PLEC) and rs7529220 (HSPG2) should be reassessed, as the observed odds ratios do not robustly categorize them as high-risk factors for Alzheimer’s disease or atrial fibrillation.

Reply 1.8: We appreciate your notice. In the revised manuscript, we present the odds ratios (OR) and p-values from the MTAG analysis, which show that *HSPG2* is associated with both traits at a GWS level, while *PLEC* is associated with AD at a GWS level and associated with AF ($P < 5 \times 10^{-6}$). Indeed, the reported ORs are small (e.g. OR=1.02 for *PLEC*); however, it is not uncommon in GWAS studies to observe associations of variants with small effect sizes, which nonetheless can have significant biological importance. In support, the genetic colocalisation as well as other subsequent analyses consistently showed that these loci are associated with both AD and AF with the same causal variant, suggesting a pleiotropic effect. Figure 3 in our manuscript demonstrates a signal for both traits at the *PLEC* locus, underscoring the shared genetic architecture.

1.9 The relevance of the identified loci to other cardiovascular traits should be discussed. If no correlation exists, the study should narrow its focus to the relationship between AD and atrial fibrillation, avoiding unnecessary data and analyses.

Reply 1.9: Thank you for your comment. As previously mentioned, we compared our findings across diverse CV traits that is thoroughly discussed in the revised version of our manuscript.

We aimed to look at several CV conditions in pairwise comparisons to explore the different pathways that may link different manifestations of AD with CV traits. For example, atrial fibrillation and coronary heart disease have different pathophysiology and therefore their cooccurrence with AD may be due to different shared mechanisms. Our study involved a series of sequential analyses, with each phase building upon the findings of the previous one to avoid unnecessary data and analyses. For instance, the colocalisation analysis was performed only for significant MTAG findings and identified 53 variant-gene-tissue associations with only four different CV traits involving 43 unique genes. We subsequently prioritised additional analyses for the indicated genes. Our gene expression analysis provided evidence of differential expression only for *PLEC* and *C1Q*, prompting further investigation into their pathways and interactions with other proteins.

We have revised our manuscript to provide clearer explanations of our methodology and findings (See reply to 1.2).

1.10 Finally, while the single-nuclei transcriptome and protein-protein interaction analyses involving *PLEC*, *NDUFS3*, and *C1Q* on AD brain tissues are noted, the study fails to elucidate a clear pathogenic connection of these targets to Alzheimer's disease and atrial fibrillation.

Reply 1.10: Thank you for your comment. We have provided a comprehensive set of analyses to draw conclusion on the potential mechanisms linking the highlighted genes with AD and AF. For example, we discuss the functions of *PLEC* and *C1Q*, aiming to elucidate their roles in the pathogenicity of AD and AF. Our study suggests that *PLEC*, a critical structural protein in the cytoskeleton, and its interactions with intermediate filaments may contribute to dysfunctions in astrocytes, potentially increasing the risk of AD. Similarly, these interactions in

cardiomyocytes could lead to dysfunctions associated with heart failure, thereby linking *PLEC* to both AD and CV disorders. Additionally, our findings suggest that higher expression of *C1Q* may enhance activity within the complement system, potentially leading to synapse loss in early AD and contributing to the formation of atherosclerotic plaques and elevated inflammatory responses in cardiomyocytes. These insights are detailed in our discussion and we have revised our manuscript to make this clearer.

“Plectin is highly expressed in the central nervous system, especially at the interfaces between glia and pial cells and between glia and endothelial cells, and is thought to be important to blood-brain barrier and pial surface integrity. Plectin deficiency in mice has been associated with diminished learning capabilities and reduced long-term memory compared to wild-type littermates. Here we provide evidence that the risk of AD may be affected via functions of plectin in astrocytes. Astrocytes play multiple roles, central to the pathology of AD, including metabolic support for neurons, modulation of brain microvascular function and, through activities associated with those of microglia, inflammatory responses. We hypothesise that these functional roles are mediated in part by interactions of plectin with intermediate filaments (IFs), microtubules and actin filaments.” (p.10, lines 232-241)

*“The role of *PLEC* in AF has been largely hypothesised to act via structural effects on the heart and cause electrophysiological abnormalities. Here, we also show evidence for upregulation of *PLEC* in cardiomyocytes of HF patients. Therefore, in accordance with the hypothesised mechanisms linking *PLEC* to AD above, *PLEC* may play related roles in cardiomyocytes for assembling and mobilising the intermediate filaments and their networks. These effects further modulate contractile function in cardiomyocytes and inflammatory responses in macrophages which may further contribute to AF.”* (p.11, lines 245-251)

*“The complement system plays a central role in synaptic remodelling in the brain and in cellular damage response more generally in the body. We hypothesise that greater expression of *C1Q* may lead to higher activity of the complement system which in turn may potentiate synapse loss in early AD. Similarly, *C1Q* has roles in the genesis of atherosclerotic plaques and in the regulation of early stages of inflammatory responses to the cardiomyocyte injury associated with a range of cardiac traits.”* (p.11, lines 258-263)

Additionally, we provide a schematic overview illustrating the pathogenic connection suggested by our study (Figure 1).

[figure redacted]

However, it is important to note that epidemiological data often face limitations in elucidating clear pathogenic pathways. While we provided additional mechanistic evidence, the precise biological mechanisms cannot be inferred. Consequently, additional research approaches are essential to complement epidemiological findings and provide deeper insights into disease pathogenesis.

We added this as a limitation:

“Finally, epidemiological data frequently encounter challenges in clarifying pathogenic mechanisms. Although we provided evidence from mechanistic experiments, the exact biological processes involved cannot be inferred. Further work is needed to validate our findings and the suggested disease pathways.” (p.13-14, lines 314-317)

Reviewer #2 (Remarks to the Author):

Overview

The work uses MTAG to examine AD with cardiovascular traits. The motivation is the long-standing association between cardiovascular disease and AD, and the method harnesses the genetic correlation to improve power for detecting genetic associations. The authors use eQTLs identified from a range of tissues from the GTEx project plus their results to perform colocalization analyses and hone in on two sites, and follow up some of those findings with analysis of gene expression. Overall, the manuscript was clear and interesting. The methods are well-described and approach seems sound, but it looked like there might be a disagreement between figure 3 and text and figure 2 does not share enough detail so it cannot be used to check against the text, which were my most important concerns. I had some specific comments below but I also wondered 1) if the authors could examine genetic correlation between AD and the traits in addition to sharing the MTAG results; and, 2) whether the authors considered testing colocalization with multiple traits at loci that seemed like they had ≥ 2 signal with several cardiovascular traits.

Reply: Thank you for your positive feedback on the clarity and interest of our manuscript and for your suggestions. We apologise for any confusion. We have carefully revised both the text and figures to ensure consistency and clarity throughout. We addressed each of your specific concerns and provided additional details to clarify any discrepancies you noted.

We provide the genetic correlation results in Supplementary Table 1.

Regarding the comment on colocalisation of multiple traits, we only performed bivariate (pairwise) MTAG analysis between AD and CV traits. Subsequently, this study design did not allow us to perform colocalisation across multiple CV traits as the MTAG AD GWAS results were different from each bivariate MTAG analysis. We opted for bivariate analysis to explore different pathways that may link different CV traits with AD (e.g. different shared pathways linking AF to AD and CAD to AD). However, in the case of systolic and diastolic blood pressure, as those two traits are correlated and reflect similar pathways, we analysed the traits in a single MTAG between AD, systolic and diastolic blood pressure and the subsequent

colocalisation was performed across these three traits. Additionally, our results did not highlight the same loci with evidence for colocalisation between different AD-CV pairs. The regions with evidence of colocalisation with multiple CV traits in Figure 2 (chromosomes 8 and 10), were associated with CV traits through different and independent signals ($R^2 < 0.1$). Thus, these loci likely do not affect the different CV traits through the same mechanism.

Major Comments

2.1. Title – at best, these data let you say that there is evidence consistent with what you'd observe with pleiotropy. Also, there are more genes implicated than just PLEC and C1Q, unless I missed something.

Reply 2.1: Thank you for your comment. We agree and have now changed the title of our manuscript to:

“Multi-trait association analysis reveals shared genetic loci between Alzheimer's disease and cardiovascular traits”.

2.2 Figure 1 would be helped by sharing the sample sizes for each GWAS used to let the reader understand their magnitudes.

Reply 2.2: Thank you for your suggestion. We modified Figure 1 accordingly and now show the sample sizes for each GWAS.

[figure redacted]

2.3 Figure 2 is difficult to read - can it just be made into a horizontal plot with labeled axes?

Reply 2.3: We apologise for this. Figure 2 presents the regional plots for the colocalised loci. Our intention with this plot was to highlight the significant results from our colocalisation analysis in a summarised format, believing that a circular form of the plot would do so more efficiently. We provide the suggested horizontal orientation in Supplementary Figure 7. Nevertheless, to address reviewer's concerns regarding the complexity of Figure 2, we applied a few modifications to simplify the plot. More specifically, we removed the outer circles corresponding to eQTL information, erased the coordinates from the x-axis and enlarged the text for the annotated genes. Finally, we removed the trait annotations from the plots and replaced them with a simpler legend.

We also made a few changes to the Figure legend, which now says:

“Fig. 2: Circular figure visualising regional plots on the colocalised loci between Alzheimer’s disease (AD) and cardiovascular traits (CV).

The figure presents the distribution of P-values ($-\log_{10}P$) from MTAG with inner orientation. The annotations show the mapped genes of the AD/CV top lead SNPs on the colocalised loci.”

2.4 Figure 3 seems to present different p-values compared to the text (lines 112-115) that should be checked.

Reply 2.4: Thank you for noticing this and apologies for the confusion. We modified our manuscript accordingly and now present the correct OR and p-values from the

MTAG analysis. The p-values in Figure 3 are now concordant with those mentioned in the text (p.4, lines 84-85)

2.5 Please provide manhattan and qq plots for each trait combination.

Reply 2.5: We provided Manhattan plots and QQ-plots for each trait (Supplementary Figures 1-6). In those plots, we compare MTAG results with those from the respective univariate GWAS. We have revised the results section appropriately:

“We examined the bivariate genetic correlation between AD and the examined CV traits (Supplementary Table 1) and visually illustrated the MTAG results alongside those from the original GWAS (Supplementary Fig. 1-6).” (p.2, lines 45-47)

2.6 Discussion - lines 218-221. The authors claim the results of their study disentangles the relationship between AD and AF. This is beyond what I think they can say with these data since the outcome in the Jansen paper is mainly from the UKB that infers case status using a question about whether the parent had dementia. This introduces a substantial amount of heterogeneity as to the underlying cause of dementia that ranges from cerebrovascular disease to a purely neurodegenerative cause, which preclude the kind of conclusion the authors made. Using Kunkle et al., 2018, which relies on a clinical diagnosis, might help this. Regardless of the specific cause of dementia, it's clear that there's a relationship between dementia and AF.

Reply 2.6: Thank you for your detailed and insightful comment. We also share the reviewer's view that there is a link between dementia and AF. Our study focused on AD, the most prevalent type of dementia, to identify genetic regions with pleiotropic effects that could indicate shared biological mechanisms contributing to their observed comorbidity. We utilised the largest available GWAS from Jansen et al. at the time of our analysis. This GWAS combined clinically diagnosed cases of AD with proxies (AD-by-proxy-individuals whose parents were diagnosed with dementia). The authors provide evidence that the correlation between the GWAS

with and without the ‘AD-by-proxy’ cases was high adding to the validity of this approach, especially regarding sample size improvement which is critical for AD GWAS. However, we agree that ‘AD-by-proxy’ may introduce heterogeneity in the results and highlight associations which may be driven by other types of dementia. We acknowledge this as a limitation of our study (see below). The revised text now more accurately represents the conclusions that can be drawn from our data.

“Here we provide evidence suggesting a shared genetic determinant that may contribute to the pathophysiology of AF and AD.” (p.10, lines 217-218)

“The GWAS for AD combined clinically diagnosed cases of AD with proxies (AD-by-proxy, individuals whose parents were diagnosed with AD). The correlation between the GWAS with and without the ‘AD-by-proxy’ cases was high adding to the validity of this approach. Nonetheless, the inclusion of individuals with parental AD diagnoses in the original GWAS may have introduced greater heterogeneity and increased probability of misclassification, suggesting that some of our observed associations might be misclassified or influenced by other types of dementia.” (p.13, lines 296-302)

2.7 Using gene expression from tissues that are not likely directly relevant for the pathogenesis of AD or one of the CVD traits rests on an assumption that the eQTLs in that tissue are generalizable. It could be that they are false positives in one tissue and are correctly null in other tissues.

Reply 2.7: We appreciate your comment. Emerging evidence suggests that AD is a systemic disease with widespread effects beyond the central nervous system. In this work, we aimed to further study the systemic nature of the disease by studying the interconnectedness of vascular health and neurodegeneration. We therefore opted to investigate agnostically all available tissues to also capture potential effects beyond the nervous system. Additionally, by studying all tissues, the analysis is not restricted to specific tissues with very small sample sizes (139 to 255 samples in brain tissues). Many eQTLs are correlated across different tissues, potentially adding to the generalizability of our findings.

However, we acknowledge that the possibility of false positive findings remains. To address this, we aimed to validate our findings through additional analyses on different populations using single-cell and single-nuclei data from brain tissue of AD cases and cardiac tissue from heart failure cases.

We believe that our approach is comprehensive, even though it comes with the limitations mentioned above. This consideration has been added to our manuscript, which now states:

“Considering the systemic nature of AD, which encompasses multiple pathways and various tissues, our study conducted a thorough investigation across all tissues, assuming correlations among many eQTLs across different tissues. However, it is important to acknowledge that some eQTLs from tissues not directly linked to AD or CV traits could represent false positive associations. To mitigate this, we validated our findings using single-cell and single-nuclei data obtained from brain tissue samples of AD cases and cardiac tissue samples from individuals with HF.” (p.13, lines 302-308)

Minor Comments

2.8 Consider adding the limitation that not all GWAS summary results are complete so signal that is not present in one of the dataset pairs would not be tested using MTAG.

Reply 2.8: Thank you for your suggestion. We modified the limitations as suggested. In the discussion, we mention:

“Additionally, we did not investigate a considerable portion of the genetic predisposition coming from rare variants ($MAF < 1\%$) as we excluded them from our analyses. This exclusion is a restriction of the MTAG method, in order to mitigate the risk of false-positive findings and biased results. Another limitation of using MTAG is that genetic variants that are not present in at least one GWAS dataset of each AD-CV pair were excluded from the corresponding pairwise MTAG analysis and therefore some variants tested in one pairwise MTAG may not be tested in another.” (p.12, lines 280-286)

2.9 Figure 3 – the eQTL line plot is confusing. Why not stack the two tissues and use dots? Either way, the association between the eQTL signal and other traits (i.e., AD, atrial fib) are not striking.

Reply 2.9: We apologise for the confusion initially caused by Figure 3. We have carefully revised the figure as suggested to enhance the illustration of the eQTL results.

Regarding your concern on the association between the eQTL signal and other traits, we now provide further clarification. This locus was identified through MTAG analysis between AD and AF and further colocalisation analysis yielded robust evidence of colocalisation, as detailed in our manuscript:

“The intronic variant rs11786896 (PLEC) explained the colocalisation of AD and AF with expression levels of PLEC in the cardiac left ventricle (PP = 0.99, %PP explained by SNP = 99%) and skeletal muscle (PP = 0.92, %PP explained by SNP = 98%) (Fig. 3).” (p.4, lines 91-94)

Additionally, we have conducted several post-GWAS analyses including gene expression, pathway analysis, and protein-protein interaction, providing supplementary evidence which strengthens the association of these traits with the identified locus.

2.10 Unlike CVD traits, AD has one region, the APOE locus, that has an outsized effect compared to other genetic loci. I would guess the results from APOC1 were driven by the association between AD and APOE.

Reply 2.10: Thank you for your insightful comment. Regarding our findings related to the *APOC1* locus, we acknowledge that *APOE* is one of the major genetic regions associated with AD, and it is indeed closely located to *APOC1*. Our MTAG analysis indicated a variant mapped to *APOC1* (rs438811). However, in our trait-trait colocalization analysis, another variant in *APOE* (rs429358) was highlighted as the

candidate causal variant. The evidence for colocalization was sufficient but not very strong (PP = 0.57).

It is important to note that the two aforementioned SNPs are in linkage disequilibrium (LD, $r^2 = 0.63$). Based on this, it is very likely that the observed association with *APOC1* is driven by the influence of the *APOE* locus. Additionally, the eQTL colocalization analysis did not show sufficient evidence for colocalization at this locus. Given the insufficient support for colocalization (see study design in Figure 1), we did not pursue further investigation into this association. However, we added the following to the discussion:

“The variant rs429358, located within APOE, showed evidence for colocalisation between AD and CAD. However, further investigation using eQTL could not prioritise any shared gene within the locus expressed in the examined tissues. Given APOE’s well-established role in AD and its potential involvement in other CV traits, including CAD, it is plausible that the locus affects the two traits independently through different pathways (horizontal pleiotropy) or through a shared gene expressed in tissues other than those examined in our study.” (p.11-12, lines 264-269)

2.11 Discussion - lines 204-5 (grammar/phrasing) - “rs11786896 expressed via PLEC and rs7529220 expressed via C1QA, C1QB, and C1QC” sounds odd. I’d phrase it something like rs1179686 is an eQTL for PLEC.

Reply 2.11: We rephrased the sentence as suggested:

“rs11786896 which was an eQTL for PLEC and rs7529220 which was an eQTL for C1QA, C1QB, and C1QC” (p.8, lines 184-185)

REVIEWERS' COMMENTS

Reviewer #1 (Remarks to the Author):

The authors have fully addressed reviewers' comments and the revised version is worth being accepted for publication as it is.

Reply: We thank the reviewer for suggesting the publication of our manuscript

Reviewer #2 (Remarks to the Author):

No matter how AD is commonly defined (e.g., clinically or pathologically), it often co-occurring with evidence of cerebrovascular disease yet whether this is pleiotropic or causal relationship is unknown -- although there is some work that suggests a direct link <https://pubmed.ncbi.nlm.nih.gov/35772923/>.

The current work focuses on finding shared genetic risk using MTAG, which uses genetic correlation to boost power for multi-trait gwas.

The most interesting portion of the work is the MTAG results along with the SMR analysis.

By contrast the differential expression and network-based analysis was descriptive without addressing the most interesting question of the work -- the shared risk for CV traits and AD dementia.

Overall, the revised manuscript nicely addressed my comments.

Minor Comments

1. line 297 - AD-proxy status was inferred based on whether the parent was diagnosed with dementia (not AD per se).

Reply: We thank the reviewer for the comment. We have clarified this distinction in the revised manuscript as suggested:

“The GWAS for AD combined clinically diagnosed cases of AD with AD-proxy status (inferred based on whether the parent was diagnosed with dementia).” (p.13, lines 296-298)